# Correction of the Splicing Defect Caused by a Recurrent Variant in *ABCA4* (c.769-784C>T) That Underlies Stargardt Disease

**DOI:** 10.3390/cells11243947

**Published:** 2022-12-07

**Authors:** Tomasz Z. Tomkiewicz, Sara E. Nieuwenhuis, Frans P. M. Cremers, Alejandro Garanto, Rob W. J. Collin

**Affiliations:** 1Department of Human Genetics and Donders Institute for Brain, Cognition and Behaviour, Radboud University Medical Center, 6525 GA Nijmegen, The Netherlands; 2Departments of Pediatrics, Amalia Children’s Hospital, Human Genetics and Radboud Institute for Molecular Life Sciences, Radboud University Medical Center, 6525 GA Nijmegen, The Netherlands

**Keywords:** antisense oligonucleotides, *ABCA4*, Stargardt disease, inherited retinal disease, splicing modulation, RNA therapy, deep-intronic, HEK293T cells, patient-derived fibroblast cells, patient-derived photoreceptor precursor cells

## Abstract

Stargardt disease is an inherited retinal disease caused by biallelic mutations in the *ABCA4* gene, many of which affect *ABCA4* splicing. In this study, nine antisense oligonucleotides (AONs) were designed to correct pseudoexon (PE) inclusion caused by a recurrent deep-intronic variant in *ABCA4* (c.769-784C>T). First, the ability of AONs to skip the PE from the final *ABCA4* mRNA transcript was assessed in two cellular models carrying the c.769-784C>T variant: a midigene assay using HEK293T cells and patient-derived fibroblasts. Based on the splicing-correcting ability of each individual AON, the three most efficacious AONs targeting independent regions of the PE were selected for a final assessment in photoreceptor precursor cells (PPCs). The final analysis in the PPC model confirmed high efficacy of AON2, -5, and -7 in promoting PE exclusion. Among the three AONs, AON2 is chosen as the lead candidate for further optimization, hereby showcasing the high potential of AONs to correct aberrant splicing events driven by deep-intronic variants.

## 1. Introduction

Inherited retinal diseases (IRDs) is an umbrella term encapsulating a spectrum of genetic disorders characterized by vision loss that can lead to blindness. Biallelic variants in the ATP-binding cassette subfamily A member 4 (*ABCA4*, MIM 601691) gene give rise to *ABCA4*-associated retinopathies of which Stargardt disease (STGD1, MIM 248200) is the most common, with an estimated incidence between 1 in 8,000 and 1 in 10,000 [1,2,3,4,5]. In STGD1, the degradation of visual acuity is described as bilateral and central due to initial death of cone photoreceptors. STGD1 is an autosomal recessive condition often emerging in juvenile persons; however, a late-onset STGD1 phenotype occurs as well [6]. In time, central vision loss can spread to complete blindness as a result of extended atrophy of retinal tissue [7]. 

In-depth *ABCA4* sequencing studies revealed that inheritance of STGD1 is a complex process in which the combination of pathogenic alleles has a strong effect on the disease onset and severity [8,9,10]. Advances in next generation sequencing techniques lead to the discovery that a significant proportion of disease-contributing variants is located outside of the coding regions of *ABCA4* [11,12]. Currently, it is estimated that deep-intronic variants represent approximately 10% of all pathological variants in *ABCA4* [13]. Among these is c.769-784C>T, a recurrent deep-intronic variant that causes the insertion of a pseudoexon (PE) in intron 6. The C to T substitution is predicted to increase the strength of a cryptic splice acceptor site (SAS), leading to recognition and insertion of a 162-nt PE into the *ABCA4* transcript and as a consequence, introduction of a premature stop codon (p.[=,Leu257Aspfs*3]). The degree to which the total mRNA transcript is affected by this PE depends on the cell model used. The percentage of total mRNA affected has been reported to be as low as 8% based on RT-PCR amplification of mutated mRNA from HEK293T cells transfected with an *ABCA4* splice reporter vector (from here on referred to as midigene) carrying the c.769-784C>T variant [12]. However, when the splice assay was performed in patient-derived photoreceptor precursor cells (PPCs), it showed that 30% of the mRNA derived from the c.769-784C>T allele contains the PE inclusion [14]. 

The relatively high prevalence of c.769-784C>T combined with the pathomolecular mechanism makes it a suitable target for antisense oligonucleotide (AON)-based splicing correction. AONs are short synthetic molecules complementary to their target mRNA. AONs modulate the molecular target either by induction of RNase H-mediated mRNA degradation, translation arrest, or splicing modulation by binding to splice sites or to exonic or intronic inclusion signals resulting in skipping or inclusion of the targeted exon [15]. AONs have an excellent performance record when used against PEs in *CEP290* [16,17], *USH2A* [18], *CHM* [19], and *OPA1* [20]. The *ABCA4* gene is a rich source of splicing-affecting, deep-intronic variants. Of these, several already have been targeted with splice-switching AONs [12,21,22,23,24], although none of the AONs designed against deep-intronic variants in *ABCA4* has entered the clinical phase of testing yet.

In this study, we expanded on the work conducted by Sangermano, Garanto et al. [12] where three AONs with 2′-O-methyl ribose modification (2-OMe) and phosphorothioate (PS) backbone were tested in HEK293T cells transfected with a midigene and in patient-derived fibroblast, both carrying the *ABCA4* c.769-784C>T variant. Those results were the basis for a more detailed investigation to identify the most potent AON(s) in PE exclusion. Additional AONs were tested in three different cell systems, and one AON was selected as the most potent for a potential therapeutic development. The selected AON candidate is consistent with the already published results where its sequence has been proven to completely restore correct splicing in the HEK293T-midigene system and patient-derived fibroblasts [12].

## 2. Materials and Methods

### 2.1. Antisense Oligonucleotide Design

The AONs were designed, as described previously [25]. The template RNA sequence structure, folding and open, and closed and partially open regions were studied using the Mfold software [26]. Moreover, we identified the serine and arginine rich splicing factor 2 proteins (SC35) splice enhancer motifs within PE using th eESE finder algorithm [27,28]. All sequences and properties of the different AONs are described in Table 1. All AONs and SON were modified with 2′-O-methoxyethyl (2′-MOE) ribose chemistry and phosphorothioate (PS) backbone chemistry. All oligonucleotides were purchased from Eurogentec and reconstituted in 1× PBS (autoclaved twice) to a final concentration of 100 µM.

### 2.2. In Vitro Rescue Studies in HEK293T Cells Using Midigenes and Antisense Oligonucleotides

Human embryonic kidney 293T (HEK293T, ATCC# CRL-3216™) cells were cultured in Dulbecco’s Modified Eagle Medium (DMEM) supplemented with 10% fetal calf serum (FCS), 1% penicillin–streptomycin, and 1% sodium pyruvate at 37 °C and 5% CO_2_. To assess the efficacy of the AONs, HEK293T cells were seeded in six-well plates at a confluence of 70%. Cells were transfected with 1.2 µg of the pDEST BA6 construct carrying the *ABCA4* mutant *ABCA4* c.769-784C>T sequence. A non-transfected, endogenous control was included. After overnight incubation, cells were trypsinized and divided into 13 wells using two 12-well plates. Next, AONs and SON were transfected at 0.5 µM to the HEK293T cells. The transfection protocol was followed, as described elsewhere [25]. Forty-eight hours after AON transfection, cells were harvested for RT-PCR transcript analysis. All experiments were performed in two independent replicates.

### 2.3. Rescue Studies Using Antisense Oligonucleotides in Fibroblasts

Fibroblast cell lines derived from individuals with *ABCA4*-associated STGD1 and healthy controls were cultured in DMEM, supplemented with 20% FCS, 1% penicillin–streptomycin and 1% sodium pyruvate at 37 °C and 5% CO_2_. The STGD1 patient-derived fibroblasts carried c.[769-784C>T;5603A>T] (p.[=,Leu257Aspfs*3;Asn1868Ile]) on allele 1 and c.1822T>A (p.(Phe608Ile)) on allele 2, whereas the healthy control carried an *ABCA4* WT gene. To assess the efficacy of the AONs, fibroblast cells were seeded in six-well plates at 0.35 × 10^6^ cells/well. After overnight incubation, the cells were transfected with 0.5 µM of the corresponding AONs or SON. Five to six hours after AON transfection an additional 1 mL of the medium was added to all wells. Fibroblasts were incubated for 24 h. After 24 h, cycloheximide (CHX) (cat. no. C4859, Sigma Aldrich, Saint Louis, MO, USA) (final concentration at 0.1 mg/mL) was used to suppress the nonsense-mediated decay (NMD) pathway and to visualize the transcript containing the PE. The transfection protocol was followed, as described elsewhere [25]. Forty-eight hours post-AON transfection, cells were harvested for transcriptional analysis by RT-PCR. All experiments were performed in at least two independent replicates. 

### 2.4. iPSC Generation and Differentiation to Photoreceptor Precursor Cells

Control (iPS15-00001) and patient (iPS15-00021) iPSCs were reprogrammed and characterized by the Stem Cell Technology Center (Radboudumc, Nijmegen, NL, USA), as previously described [29]. iPSCs were differentiated to PPCs following a 30-day 2D differentiation protocol [30]. iPSCs were dissociated with ReLeSR (cat. no. 05872, Stemcell Technologies, Vancouver, Canada) and counted. A total of 0.7 × 10^6^ cells were seeded per well on a 12-well plate coated with Matrigel (cat. no. 354230, Corning, Tewksbury, MA, USA). Seeded iPSCs were cultured with Essential-Flex E8 medium until 100% confluency was reached. At this point the Essential-Flex E8 medium was changed for differentiation medium (Cl). The CI medium consisted of DMEM/F12 supplemented with nonessential amino acids (NEAA, Sigma Aldrich, Saint Louis, MO, USA), B27 supplements (cat. no. 12587010, Thermo Fisher Scientific, Bleiswijk, NL), N2 supplements (cat. no. 17502048, Thermo Fisher Scientific, Bleiswijk, NL), 100 ng/µL of insulin growth factor-1 (IGF-1, cat. no. I3769-50UG, Sigma Aldrich, Saint Louis, MO, USA), 10 ng/µL of recombinant fibroblast growth factor basic (bFGF, cat. no. F0291, Sigma Aldrich, Saint Louis, MO, USA), 10 µg/µL of heparin (cat. no. H3149-25KU, Sigma Aldrich, Saint Louis, MO, USA), 200 µg/mL of recombinant human COCO protein (cat. no. 3047-CC-050, R&D Systems, Minneapolis, MN, USA), and 100 µg/mL of primocin (cat. no. ANT-PM-2, Invivogen, Toulouse, France). In each well, half of the medium was replaced daily. AONs were delivered on day 28 of differentiation, CHX was added to the medium on day 29, and PPCs were harvested on day 30.

### 2.5. Rescue Studies Using Antisense Oligonucleotides in Photoreceptor Precursor Cells

On day 28 of differentiation, PPCs were treated with AONs (0.5 µM and 1 µM) by mixing the AONs directly with the differentiation medium. After 24 h of incubation CHX was added at a final concentration of 0.1 mg/µL and incubated overnight. Forty-eight hours after AON delivery, the cells were ready for collection. Medium was collected from all wells and centrifuged, and the cell pellet was used for RNA isolation. 

### 2.6. RNA Isolation and cDNA Synthesis

RNA extracted from HEK293T, fibroblasts, and PPC was isolated using the Nucleospin RNA kit (Machery-Nagel, Düren, Germany) following the manufacturer’s instructions. For HEK293T cells, 1 µg of total RNA was used for cDNA synthesis using the iScript cDNA Synthesis kit (Bio-Rad, Hercules, CA, USA) according to the provider’s protocol. For fibroblasts and PPCs, 1 µg of total RNA was used for all cDNA synthesis reactions using the SuperScript VILO Master Mix (cat. no. 11755050, Thermo Fisher Scientific, Carlsbad, CA, USA) and following the manufacturer’s instructions.

### 2.7. iPSC and PPC Characterization

RNA isolation and cDNA synthesis were performed on undifferentiated iPSC and differentiated PPCs, as described above. Primers for the quantitative PCR analysis are listed in Appendix A. The qPCR was setup with GoTaq Real-Time qPCR Master Kit (Promega, Madison, WI, USA). The reactions were set up in triplicates and processed using the 7900HT fast real-time PCR system. The housekeeping GUSB gene was used for normalization, and the relative quantification was based on the 2^−(ΔΔCt)^ method [31].

### 2.8. ABCA4 Transcript Analysis

RT-PCR was used to assess the splicing correction in all three models. For the HEK293T-midigene model, primers were located in *RHO* exon 3 (forward) and *ABCA4* exon 7 (reverse). Exon 5 of *RHO* was used to assess the transfection efficiency of the midigene constructs. For the RT-PCR analysis of splicing in patient-derived fibroblasts and patient-derived PPC models, primers were located in exon 5 (forward) and exon 7 (reverse) of *ABCA4*. Actin (*ACTB*) primers were used as a control. Primer sequences are listed in Appendix A.

For the HEK293T-midigene model, 40 ng of cDNA was used for all the *ABCA4* reactions and for the *RHO* reaction. All reaction mixtures (25 μL) contained 10 μM of each primer pair, Taq DNA Polymerase 1 U/μL (Roche, Mannheim, Germany), 1× PCR buffer with MgCl_2_, 1.5 mM MgCl_2_, 2 mM dNTPs, and 40 ng cDNA. PCR conditions for the *RHO*-*ABCA4* midigene fragment from *RHO* exon 3 to *ABCA4* exon 7 were as follows: 94 °C for 3 min, 35 cycles of 30 s at 94 °C, 30 s at 58 °C, and 90 s at 72 °C, followed by a final step of 2 min at 72 °C. *RHO* PCR on exon 5 was performed under the same conditions except for an elongation time of 30 s. Ten µL of the *RHO*-*ABCA4* PCR product and 10 μL of the *RHO* amplicon were resolved on a 2% (*w*/*v*) agarose gel.

For patient-derived fibroblasts and patient-derived PPCs, 80 ng of cDNA was used for all the *ABCA4* reactions and 40 ng for the *ACTB* analysis. All reaction mixtures (25 μL) contained 10 μM of each primer pair, Taq DNA Polymerase 1 U/μL (Roche, Mannheim, Germany), 1× PCR buffer with MgCl_2_, 1.5 mM MgCl_2_, 2 μM dNTPs, and 80 or 40 ng cDNA. PCR conditions for *ABCA4* fragments from exons 5 to 7 were as follows: 94 °C for 3 min, 35 cycles of 30 s at 94 °C, 30 s at 58 °C, and 90 s at 72 °C, followed by a final step of 2 min at 72 °C. PCR was performed in the same conditions, and 25 μL of the *ABCA4* PCR product and 10 μL of the actin amplicon were resolved on a 2% (*w*/*v*) agarose gel.

Semi-quantitative analysis of the bands was performed by using Fiji software (software version no. 2.9.0, Bethesda, MD, USA) [32] (Appendix A). Percentage (%) of splicing correction was calculated from the decrease of aberrant transcript by the AON treatment relative to the corresponding non-treated mutant condition.

### 2.9. Statistical Analysis

Data were represented as means ± SD and analyzed with GraphPad Prism 9 software (GraphPad, San Diego, CA, USA). To study the differences between treated and untreated conditions, we implemented the one-way ANOVA test. *p*-values smaller than 0.05 were considered statistically significant.

## 3. Results

### 3.1. AONs Are Capable of Rescuing the Splicing Defect in Midigene Splicing Assays

The c.769-784C>T variant strengthens a pre-existing splice acceptor site in intron 6 that leads to the insertion of the PE, as depicted in Figure 1A. To prevent this PE from being recognized by the splicing machinery and on top of the three AONs that were used previously [12], six novel AONs were designed (characteristics in Table 1) covering different regions within the PE (Figure 1B). Upon co-transfection of the nine AONs together with the mutant midigene, we observed that the PE-containing transcript fragment without AON treatment represented 19.5% of the total transcript. Almost all AONs reached 100% splicing correction, as depicted in Figure 2A, and the RT-PCR results were confirmed with Sanger sequencing (Appendix A). Only AON3, -4, and -8 were short of complete PE exclusion (7%, 8%, and 5% PE inclusion, respectively). Transfection of SON did not have any effect on the PE. The high success rate in PE exclusion, even with the AONs that failed to reach statistical significance, prompted us to re-test the same AONs in patient-derived fibroblasts.

### 3.2. AON2, -5, and -7 Are the Most Effective None-Overlapping AONs in Splicing Correction in Patient-Derived Fibroblasts

To study the potential of these AONs in patient-derived cells, fibroblasts were used. The same nine AONs were transfected into patient-derived fibroblasts that were then treated with CHX to inhibit the NMD pathway. This allowed for visualization of all the transcripts and accurate assessment of the ratio between correct transcript and PE transcript. In control fibroblasts, the PE represents ~16% of the total transcript. In fibroblasts carrying the c.769-784C>T variant, the PE transcript represented ~69% of the total transcript. After the AONs were transfected, we observed a various effectiveness on splicing correction, as captured by gel image and semi-quantification in Figure 2B and Appendix A. The identity of the RT-PCR amplicons was confirmed with Sanger sequencing (Appendix A). As all samples were equally treated and analyzed simultaneously, we could compare the efficacies based on the semi-quantification analyses. The SON was also transfected and had no effect on PE. AON1, -4, and -6 targeting the cryptic SAS had the lowest splicing-correction efficacies and did not reach statistical significance (*p* = 0.70, *p* = 0.29, and *p* = 0.30, respectively, when compared to AON-untreated, patient-derived cycloheximide control). Additionally, an AON-artifact was observed with AON1, which has been already described previously [10]. Moreover, the same AON-artifact was observed with AON3 (Appendix A). For AON2, -3, -5, -8, and -9, the PE exclusion reached statistical significance when compared to the AON-untreated, patient-derived CHX control (*p* < 0.05). AON7 did not reach statistically significant splicing correction (*p* = 0.0596); however, it was the only AON targeting a strong exonic splicing enhancer (ESE) motif located in the PE, as shown in Figure 1B. Among the AONs targeting the cryptic SDS, AON5 was the most efficacious in that group.

From the pool of tested AONs, we selected AON2, -5, and -7 for screening in PPCs. The overlapping target sequence of AON2 and AON9 with strong ESE motif, as shown in Figure 1B, and overall better performance of AON2 led to AON9 exclusion from the final efficacy assessment in patient-derived PPCs. The selected three AONs also had no detrimental effect on splicing in fibroblasts carrying *ABCA4* wild-type sequence (Appendix A). 

### 3.3. AON2, -5, and -7 Are Highly Efficacious in Correcting PE in Patient-Derived PPCs

Patient-derived iPSCs were differentiated into PPCs using a 30-day long protocol. The successful differentiation of the cells was assessed by qPCR. The results showed an increase of expression of retina-specific genes and emphasized *ABCA4* expression compared to day 0 iPSCs (Appendix A). Similar to the patient-derived fibroblasts, the PE was visualized through the inhibition of the NMD pathway via CHX treatment. AON-driven splicing correction was achieved by gymnotically delivering AONs at day 28 of differentiation at two concentrations, 0.5 µM and 1 µM. Both *ABCA4* control and c.769-784C>T cell lines were exposed to the treatment. SON was delivered simultaneously with the AONs at 1 µM in the same manner. The RT-PCR results were confirmed with Sanger sequencing (Appendix A). As observed in the two previous experiments, a small degree of PE inclusion was found in the CHX-treated line expressing *ABCA4* wild-type sequence (5% PE inclusion). PE insertion in the CHX-treated mutant line represented 34% of the total transcript. Statistically significant PE exclusion was achieved with all AONs at both concentrations (*p* < 0.05). AON2 delivered at 0.5 µM and AON7 delivered at 0.5 µM and 1 µM were the most potent, whereas AON5 delivered at 0.5 µM was the least efficacious (Figure 3). Inter-efficacy comparison of AON2 at 0.5 µM and 1 µM showed that AON2 performed slightly better at the lower concentration, while the same analysis for AON7 showed the difference in efficacy between the two concentrations is negligible. In the control PPC line, the residual expression of the PE was detected by RT-PCR and semi-quantified using Fiji software. In the not-treated control line, the PE represented 5%. After the AON treatment, the PE expression represented between 3% to 5% (Appendix A) of the total transcript. Splicing aberrations in the control PPC line due to exposure to any of the AONs were not observed.

The three-tier screen approach identified AON2, -5, and -7 as the most promising AONs. The initial selection of these three was driven by the results obtained from patient-derived fibroblasts. The assumed efficacy of the selected three AONs was confirmed in the patient-derived PPC. Testing of AONs in the more retina-like environment of the PPC model allowed for more precise efficacy assessment leading to categorization of AONs potency in favor of AON2 due to its consistency in performance across the used cell models.

## 4. Discussion

In this study we aimed at identifying the most potent AON to be further developed for AON-based therapy for STGD1 caused by c.769-784C>T. The HEK293T-midigene system did not allow for distinguishing any AON for further testing; hence, the same AON set was tested in the patient-derived fibroblasts. Of the AONs targeting PE SAS, AON1, -4, and -6 were the least effective and did not reach statistically significant splicing correction values. From the pool of AONs located at the cryptic SDS, AON5 was the most efficacious. In addition, of the individual AONs targeting splicing enhancing motifs, AON2 and -7, proved to be effective for PE exclusion. We chose AON2 as the lead therapeutic candidate. Efficacy of AON2 has already been demonstrated in previous work and further expanded here [12]. The target location of the AON agrees with our previous observation on the relevance of targeting ESE motifs for effective PE exclusion. The *ABCA4* cDNA size (7 kb in size) and the extremely high allelic heterogeneity (over thousand causative *ABCA4* variants identified to date, www.lovd.nl/ABCA4, accessed on 20 October 2022 [10]) have made it difficult to develop an effective and universal treatment for STGD1. Fortunately, using AON technology, it is possible to address the significant proportion of deep-intronic, splicing-affecting *ABCA4* variants. Here, we showed that AON2 efficacy observed in HEK293T-midigene and patient-derived fibroblasts models was replicated in a PPC model, and its splicing rescue potential is not dependent on a single concentration.

In this study, we used 2′-MOE and the PS backbone modification. Historically, the combination of 2′-OMe and PS modifications yielded optimistic results in splicing correction in IRDs contributing genes by facilitating resistance to nucleases and increased target binding [12,18,19,21,22,23,33,34,35,36]. Following the rapid developments in the area of AON modifications, the 2′-MOE group has been steadily replacing 2′-OMe in in vitro and clinical use. In addition to the even greater resistance to endonucleases and improved on-target binding, 2′-MOE was observed to have a low toxic profile [37,38]. In the context of this study, the most convincing argument for using 2′-MOE chemistry in in vitro AON selection and optimization is the record of improvements in the splicing-switching capacity of AONs carrying 2′-MOE-PS [39]. The design of the tested AONs attempted to cover the most significant features participating in the splicing process; these include SAS and SDS as well as ESE motifs [25], in particular those for SC35 that are located within the PE. Our findings among the nine AONs allowed us to exclude six AONs, and the potencies of the remaining three were further scrutinized in a PPC model at 0.5 and 1 µM concentrations. All were able to promote PE exclusion with AON2 taking the lead.

The efficacy of the designed AONs was assessed in a three-tier cell system. The HEK293T-midigene system and patient-derived fibroblasts were used to facilitate early identification of the most effective AONs. In the HEK293T-midigene system almost all AONs reached 100% splicing correction. Due to the observed high efficacy, all AONs were tested in patient-derived fibroblasts. We speculate that the poor selection ability displayed by the HEK293T-midigene system is due to the limited genetic content of the *ABCA4* gene against which the AONs were tested. The midigene used here contained parts of intron 6 and exon 7 only, as depicted in Figure 2A. The lack of sufficient or the entire genetic content has been a recognizable, inherent limitation of midigenes. The ability of AONs to mediate its splice-switching effect depends on many factors, one of them being the accessibility to the target region, which in turn depends on the RNA folding. We hypothesize the *ABCA4* fragment present in the midigene drastically simplifies the complexity of the 3D folding, therefore making it easier for the AON to exert its effect. Despite these limitations, the HEK293T-midigene system is still considered useful because it allows for performing splice assays on hypothetical variants in genes with restricted expression for which a patients’ biopsy is not available, and as depicted here and in other studies, it allows for mass screening of AONs [24]. The advantage of fibroblasts over the HEK293T-midigene system is the endogenous expression of *ABCA4*, which resolves the problem of simplified RNA folding. The result of added genetic complexity revealed AON1, -4, and -6 failed to reach statistically significant splicing correction and therefore, were excluded from further studies. Another important feature observed in the fibroblasts model used was the exaggerated effect of the variant on splicing. Looking at the rate of PE containing transcript to WT transcript in fibroblasts, it is clear over 50% of transcript is faulty (Figure 2B). This extreme ratio was not observed in PPCs whose gene expression profile resembles that of retinal tissue more than fibroblasts. Exaggeration of such results could lead to a biased interpretation of variant severity if tested alone. The PPC model was the final model in which the efficacy of the selected AONs was assessed. The advantage of this model is its similarity to retina-like cells, allowing to replicate retina-specific splicing dynamics [21,40,41]. The retina-like nature of the cells was confirmed by qPCR demonstrating the relative increase in gene expression associated with retinal tissue and a decrease in expression of genes related to pluripotency.

STGD1 and *ABCA4* have been researched extremely thoroughly leading to initiation of multiple clinical trials that can be generally divided into cell transplantation therapies, compound therapies, and gene augmentation therapies. AON-based therapies for STGD1, however, have yet to reach clinical trial phase [7]. In the age of personalized medicine, AONs can be used to modify ‘more common’ STGD1-contributing variants, such as c.769-784C>T, and to treat the ultra-rare or even private variants. Splice-switching AONs have already made a mark in the field of therapies for IRD with ongoing clinical trials for Leber congenital amaurosis due to a deep-intronic mutation in *CEP290* c.2991+1655A>G [16] and retinitis pigmentosa caused by coding variants in exon 13 of *USH2A* [42]. The relatively high prevalence of STGD1 plus the high frequency of splicing affecting variants make the disease and *ABCA4* excellent candidates for a next AON-focused clinical trial. To facilitate the transition from pre-clinical testing to the clinical setting, detailed and meticulous evaluations of AONs at the early screening stages, as depicted here, ensure only the most promising candidates are considered for further work up and eventually clinical trials. To that end, we promote AON2 for the next step in therapy development for STGD1 associated with the c.769-784C>T variant. The selected AON needs detailed pre-clinical investigation to be completed prior to clinical trial commencement, including sequence optimization, conformation of protein rescue carried out in in vitro models and thoroughly investigated, hybridization-dependent and -independent toxicities carried out in appropriate animal models.

In this work, we collected strong evidence of pre-mRNA splicing correction in multiple models, including patient-derived PPCs, which paves the way for testing of the candidate lead, AON, in 3D retinal organoids. The sequence length of AON2 will be subjected to optimization to investigate possible improvements in uptake. For c.769-784C>T, quantitative analysis of protein function is challenging because ABCA4 activity is not fully abolished as it would be in a case of severe variants, such as c.768G>T or c.5461-10T>C. As of now, a robust and reliable functional assay for ABCA4 protein is not available. Possible methods to measure ABCA4 activity are biochemical tests, such as ATPase and retinoid binding assays [43]. Another likely target for indirect measure of ABCA4 activity is A2E assessment. A2E is the toxic component found in lipofuscin [44,45]. For a comprehensive overview of the protein activity and the effect of AON treatment, ABCA4 activity needs to be assessed in WT protein and in the complex heterozygote mutant with and without the variant of interest. Elucidation of the pathological threshold of ABCA4 protein activity is another important research goal that would add to the general understanding of the relationship between *ABCA4* variants and disease phenotype. So far, great effort has been placed into correlating patients’ phenotypes with genotypes [7]. Unfortunately, the pathological threshold of ABCA4 protein levels is still speculative; however, it has been postulated to be between 30% to 40% of remaining protein activity. Hybridization-dependent toxicity is another unwanted interference with off-target RNA processing, which can be minimized by BLAST analysis [25] and/or an in silico analysis with the RNAhybrid platform developed by Ionis Pharmaceuticals [46]. RNAseq is also a suitable technology to investigate the predicted off-target genes; however, using the most relevant cell phenotype is crucial as splicing dynamics vary between tissue types and can influence the final interpretation [47]. Hybridization-independent toxicity is driven by specific chemical groups added to the oligonucleotides, which has been described above. 

The c.769-784C>T variant is frequently found in cis with a coding variant i.e., c.5603A>T (p.(Asn1868Ile)), especially in Dutch cohorts, or c.5882G>A (p.(Gly1961Glu)). In both cases, those complex alleles have been associated, in general, with late-onset STGD1 [12,48]. However, the presence of additional severe variants in trans complicates the predicted severity and onset of STGD1 [12,48]. Our semi-quantification results derived from the PPC model indicated at least 30% of the total transcript is prematurely terminated due to the presence of the c.769-784C>T variant (Appendix A). With the absence of a reliable functional assay for ABCA4, it is impossible to unambiguously define to what degree the non-functional mRNA impairs the protein function; hence, it is problematic to give a clear-cut description of c.769-784C>T contribution to pathogenicity. Despite this shortcoming and given that onset of STGD1 requires a protein activity to fall below a pathogenicity level, we speculate in the absence of a reliable and readily available treatment for coding variants, almost 100% correction of aberrant splicing due to c.769-784C>T by AON treatment may contribute to slowing down the progression of STGD1 or even bring the disease progression to its halt. 

Taken together, in this study, we showed that PE insertion due to *ABCA4* c.769-784C>T variant was corrected with high efficacy at the pre-mRNA level using AONs. We recommend a detailed protein and toxicity workup to provide evidence of AON-mediated rescue of *ABCA4* activity and safety. Among the AONs developed for severe *ABCA4* variants, our AON is the first to address complex heterozygous alleles of which only one variant is suitable for correction.

## Figures and Tables

**Figure 1 cells-11-03947-f001:**
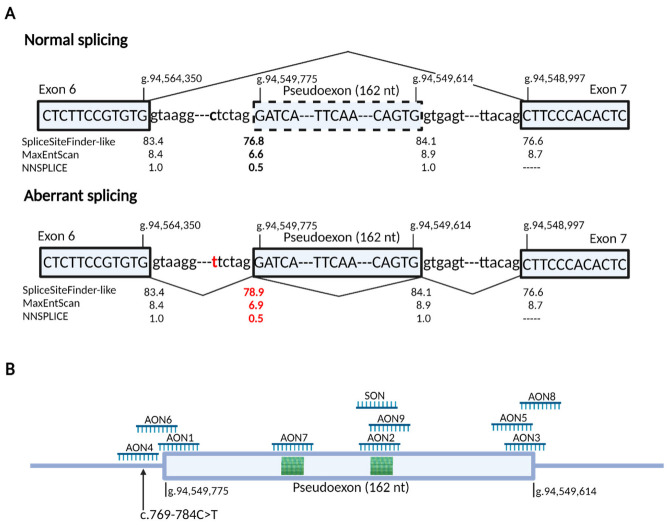
In silico characterization of the effect caused by the deep-intronic c.769-784C>T variant and schematic representation of the position of AONs. (**A**) The diagram represents boundaries of the 162-nt pseudoexon (PE) and the neighboring exons 6 and 7. The location of the c.769-784C>T variant is indicated in red. The lines illustrate the normal splicing using canonical spice sites and aberrant splicing using the cryptic splice sites due to the deep-intronic c.769-784C>T variant. The predicted values of splice site strength were acquired from the Alamut software. The C to T substitution strengthens the splice acceptor site slightly which is captured in red. (**B**) Positions of the AONs and SON targeting three distinctive sites of the PE; AON1, -4, and -6 target the cryptic splice acceptor site, AON3, -5, and -8 target the cryptic splice donor site, and AON2, -7, and -9 and SON target the predicted high score ESE motifs illustrated in green.

**Figure 2 cells-11-03947-f002:**
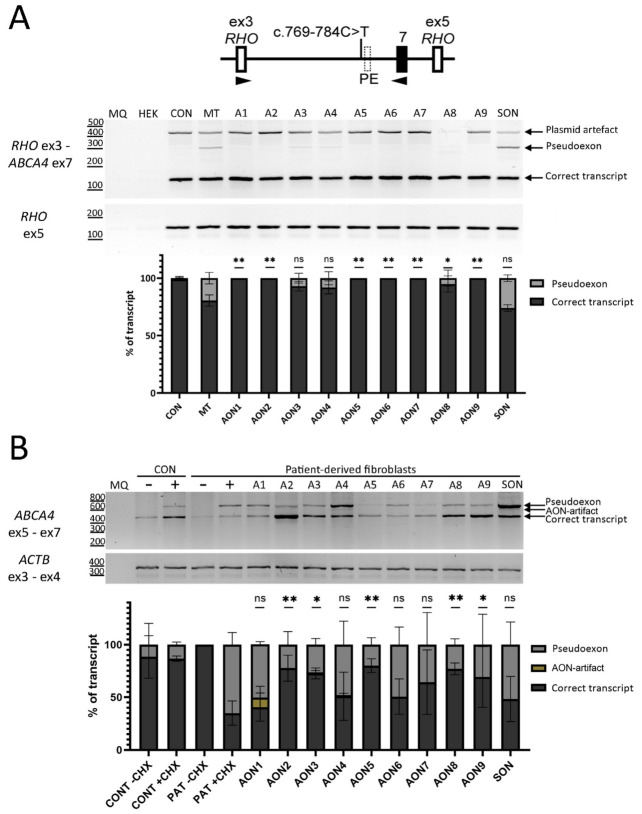
AON-based rescue for the deep-intronic c.769-784C>T variant in two cellular models. Analysis of splicing correction by RT-PCR upon AON delivery. (**A**) The midigene carrying the c.769-784C>T variant and control midigene were transfected in HEK293T except for the endogenous HEK293T expression control (HEK). The nine AONs and SON were co-transfected to cells expressing the midigene carrying c.769-784C>T variant and were labelled from A1 to SON. The AON-not transfected wells identified as control (CON) and mutant (MT) were used as a reference for splice correction. The genomic region of the used midigene is represented on top of the RT-PCR result. The graph below represents the semi-quantification of the resulting RT-PCR products, showing the percentage of correct and PE inclusion transcript. Statistically significant PE exclusion was achieved for 7/9 AONs. MQ shows the negative control of the PCR reaction, and amplification of exon 5 of *RHO* gene was used as a transfection control. Data (*n* = 2) are presented as mean ± SD (*ns*: not significant, * *p* < 0.05, ** *p* < 0.01). (**B**) The same nine AONs and SON were tested in patient-derived fibroblasts. AONs and SON were transfected to patient-derived fibroblasts carrying the c.769-784C>T variant and were labelled from A1 to SON. To control nonsense-mediated decay (NMD), cells were grown either in presence (+) or absence (−) of CHX. The AON-not transfected control (CON) and patient-derived fibroblasts were used as a reference point for splicing-correction. The graph below represents the semi-quantification of the resulting RT-PCR products, showing the percentage of correct and PE inclusion transcript. Statistically significant PE exclusion was achieved for 5/9 AONs. MQ shows the negative control of the PCR reaction and amplification of exon 3—exon 4 of *ACTB* gene was used as a loading control. Data (*n* = 3) are presented as mean ± SD and compared to the AON-untreated condition (*ns*: not significant, * *p* < 0.05, ** *p* < 0.01.

**Figure 3 cells-11-03947-f003:**
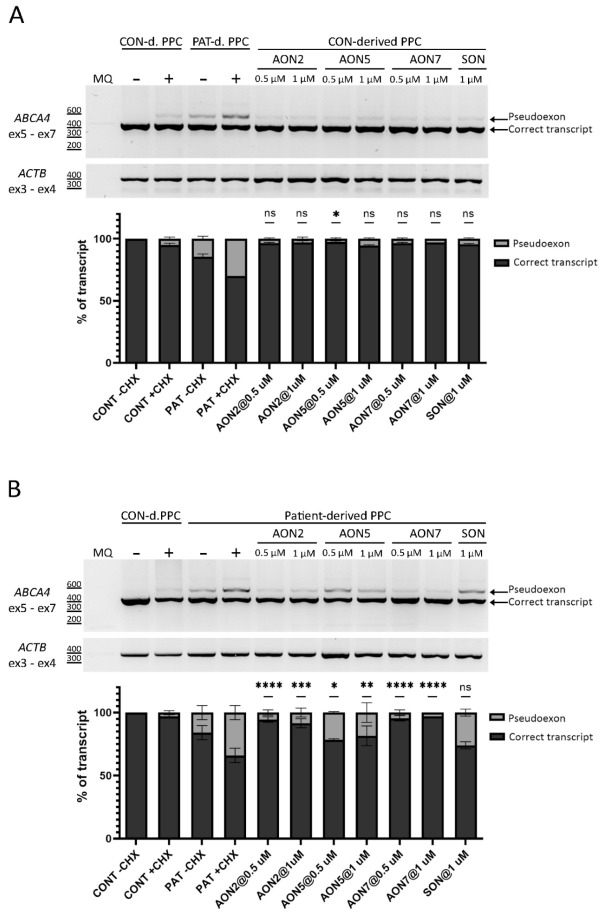
AON-based rescue for the deep-intronic c.769-784C>T variant in PPCs. Analysis of splicing correction by RT-PCR upon AON delivery to control and patient-derived PPCs. Three AONs were delivered naked to control (**A**) and patient-derived (**B**) PPCs at 0.5 µM and 1 µM concentrations. SON was delivered at 1 µM only. Cells were cultured in the absence (−) or presence (+) of CHX. The graphs below represent the semi-quantification of the RT-PCR products, showing the percentage of correct transcript and PE inclusion transcript. Statistically significant PE exclusion was achieved for none of the tested AONs in the control-derived PPCs at any concentration. MQ shows the negative control of the PCR reaction and amplification of exon 3—exon 4 of *ACTB* gene was used as a loading control. Data (*n* = 2) are presented as mean ± SD and compared to the AON-untreated condition (*ns*: not significant, * *p* < 0.05, ** *p* < 0.01, *** *p* < 0.001, **** *p* < 0.0001).

**Table 1 cells-11-03947-t001:** Antisense oligonucleotide (AON) sequences and characteristics.

Name	Sequence 5′→3′	Length (nt)	% GC	Tm (°C)
AON1	GAUGGAAUCACUGAUCCUAG	20	45	49.7
AON2	AGCUCCAGAGACUGAUGUGA	20	50	51.8
AON3	CUCACCACUGCUCCUGC	17	65	51.9
AON4	CCUAGAAGAGUCAUGUAGGA	20	45	49.7
AON5	ACCACUGCUCCUGCUUUUGC	20	55	53.8
AON6	GGAAUCACUGAUCCUAGAAGA	21	53	50.5
AON7	CUGGAAACCAAGGUCAUGUC	20	50	51.8
AON8	AUGUGAACUCACCACUGCUCC	21	52	54.4
AON9	CAAGCUCCAGAGACUGAUGU	20	50	51.8
SON	UCACAUCAGUCUCUGGAGCU	20	50	51.8

## Data Availability

Data is contained within the article or Appendix A.

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
