# Peer review of "Correction of the Splicing Defect Caused by a Recurrent Variant in ABCA4 (c.769-784C>T) That Underlies Stargardt Disease"

_cells, 2022, doi:10.3390/cells11243947_

Round 1

Reviewer 1 Report

The paper by Tomkiewicz et al tested antisense oligonucleotides with the attempt of future therapies to cure Stargardt disease. Their technique could provide innovative potential gene therapeutic interventions correct aberrant splicing events driven by deep-intronic variants. This was an interesting experimental design and they developed relevant scientific tools for gene therapy of inherited diseases. Herein I have some additional comments to further consider.

1.Introduction: please briefly describe RNA-based therapeutics  with the antisense oligonucleotide-based splicing correction in your introduction for readers, who is not familiar with the technique. Why that technique was chosen?

2.For ABCA4 retinopathies there are more recent papers published, not just after 2016, such as Saoud Al-Khuzaei et al. 2021. And if you mention, that Stargardt is the most common, please indicate incidence.

3.The last sentence of the introduction is very subjective interpretation with the word “ extremely” use, please change that and complete the conclusion of the referred study. In what context it was extremely successful in the PE exclusion?

4.The materials and methods are well described as well as the results.

5.The attempt to cure Stargardt disease is very relevant and the authors showed great results towards that. In their discussion they mention further steps, such as ABCA4 activity and off-target effect testing, which are reasonable next steps.

After the small corrections recommended above, I highly support the publication of the data.

Author Response

First of all, we would like to thank all the positive evaluations by the reviewers. Below, we have addressed the individual comments.

Reviewer 1:

The paper by Tomkiewicz et al tested antisense oligonucleotides with the attempt of future therapies to cure Stargardt disease. Their technique could provide innovative potential gene therapeutic interventions correct aberrant splicing events driven by deep-intronic variants. This was an interesting experimental design and they developed relevant scientific tools for gene therapy of inherited diseases. Herein I have some additional comments to further consider.

1.Introduction: please briefly describe RNA-based therapeutics with the antisense oligonucleotide-based splicing correction in your introduction for readers, who is not familiar with the technique. Why that technique was chosen?

A: Following the reviewer’s suggestion, we have added a short general introduction on RNA-based therapeutics, as well as an explanation why we chose this technique for ABCA4 variants in general, and the c.769-784C>T variant in particular.

‘AONs are short synthetic molecules complementary to their target mRNA. AONs modulate the molecular target either by induction of RNase H-mediated mRNA degradation, translation arrest and splicing modulation by binding to splice sites or to exonic or intronic inclusion signals resulting in skipping or inclusion of the targeted exon [15]. Among their many utilities, AONs can be used as splicing modulators for PE skipping-based therapeutic approaches [15], and have an excellent performance record while used against PEs in CEP290 [16, 17], USH2A [18], CHM [19] and OPA1 [20]. The ABCA4 gene is a rich source of splicing-affecting deep-intronic variants. Of these, several already have been targeted with splice-switching AONs [12, 21-24], although none of the AONs designed against deep-intronic variants in ABCA4 has entered clinical phase of testing yet.’

2.For ABCA4 retinopathies there are more recent papers published, not just after 2016, such as Saoud Al-Khuzaei et al. 2021. And if you mention, that Stargardt is the most common, please indicate incidence.

A: The suggested citation was added to the introduction part in line (line 38, ref 5). The rate of incidence was added in line 38 ‘, with an estimated incidence between 1 in 8,000 and 1 in 10,000’

3.The last sentence of the introduction is very subjective interpretation with the word “ extremely” use, please change that and complete the conclusion of the referred study. In what context it was extremely successful in the PE exclusion?

A: The last sentence in the introduction was corrected to be less ambiguous, line 79 to 82 ‘The selected AON candidate is consistent with the already published results, where its sequence has been proven to completely restore correct splicing in HEK293T-midigene system and patient-derived fibroblasts.’

4.The materials and methods are well described as well as the results.

A: The authors would like to thank the reviewer for her/his positive comments.

5.The attempt to cure Stargardt disease is very relevant and the authors showed great results towards that. In their discussion they mention further steps, such as ABCA4 activity and off-target effect testing, which are reasonable next steps.

After the small corrections recommended above, I highly support the publication of the data.

A: We acknowledge the overall positive comments of reviewer 1.

In the attachement please find a "Cover letter to the reviewers".

Reviewer 2 Report

This is a good article in which the authors address the ability of antisense oligonucleotides to correct the inclusion of a pseudoexon (PE) in the mRNA of a prevalent (Stargardt disease-causative) ABCA4 mutant allele. With this purpose they use three different cellular systems: HEK-293T midigene assay, patient-derived fibroblasts and photoreceptor precursor cells derived from patient iPSC. The article methodology is original, the text is marvelously written and conclusions are adequately supported by the results. Yet the authors should address the queries raised by this referee in order to improve their paper.

Major Points

The Materials and Methods are quite sophisticated, but well explained in an understandable fashion. Yet, this referee believes that this section is too long and should be shortened to less than three printed pages.

The article should greatly benefit from the authors testing adequate splicing not only at the mRNA level, but also by means of western blotting. This should allow them to assess whether correction of PE inclusion actually results in an increased amount of correct (wt) ABCA4 protein. If they can prove this in (at least) one of the three cellular systems, that should suffice.

Should the authors have any result, albeit preliminary, regarding preclinical investigation on a given rodent model or 3D retinal organoids that they could show in Results or comment on in the Discussion, this would give significant value to this paper.

Minor Points

Line 43: Do the authors mean ‘inheritance’ instead of ‘heritability’?

Line 48: Please choose between ‘~10%’ and ‘approximately 10%’.

Line 86: The term ‘co-transfected’ in the expression ‘then co-transfected with AONs’ means that after midigene the cells are then (doubly) transfected with a particular AON, or else that they were co-transfected with a set of AONs? Please clarify this ambiguity.

Line 131: Where it reads ‘Hundred microliter’ it must read ‘One hundred mL’.

Line 151: Where it reads ‘recombinant human COCO’ it must read ‘recombinant human COCO protein’.

Line 373: Please choose between ‘fibroblasts’ and ‘cells’.

Suppl. Fig. S4 legend: Please rewrite it for the sake of clarity. Ensure that all abbreviations are defined in all legends (for instance, WT, MT here and PAT, MQ in Suppl. Fig. S3).

Author Response

First of all, we would like to thank all the positive evaluations by the reviewers. Below, we have addressed the individual comments.

Reviewer 2:

This is a good article in which the authors address the ability of antisense oligonucleotides to correct the inclusion of a pseudoexon (PE) in the mRNA of a prevalent (Stargardt disease-causative) ABCA4 mutant allele. With this purpose they use three different cellular systems: HEK-293T midigene assay, patient-derived fibroblasts and photoreceptor precursor cells derived from patient iPSC. The article methodology is original, the text is marvelously written and conclusions are adequately supported by the results. Yet the authors should address the queries raised by this referee in order to improve their paper.

Major Points

The Materials and Methods are quite sophisticated, but well explained in an understandable fashion. Yet, this referee believes that this section is too long and should be shortened to less than three printed pages.

A: We have shortened the Materials and Methods section to comply with the suggestion of this reviewer.

The article should greatly benefit from the authors testing adequate splicing not only at the mRNA level, but also by means of western blotting. This should allow them to assess whether correction of PE inclusion actually results in an increased amount of correct (wt) ABCA4 protein. If they can prove this in (at least) one of the three cellular systems, that should suffice.

Should the authors have any result, albeit preliminary, regarding preclinical investigation on a given rodent model or 3D retinal organoids that they could show in Results or comment on in the Discussion, this would give significant value to this paper.

A: We agree with the reviewer that demonstrating the effect of RNA correction at the protein level would improve the paper. However, none of our model systems is suitable for this; using the HEK293T-midigene system, no protein is produced thus Western blot analysis is not possible. In patient-derived fibroblasts as well as in the photoreceptor precursor cells, detection of the ABCA4 protein by Western blotting is technically challenging, due to the overall low expression levels. In addition, as the c.769-784C>T is a hypomorhic allele with residual wild-type mRNA and protein expression, the expected difference in protein levels between treated and untreated cells is probably small, hence the limitation of these models impedes an accurate evaluation of the effect of the AONs on ABCA4 protein rescue.

Unfortunately, at the moment we are also not in possession of preliminary results regarding preclinical investigation in a rodent model or 3D retinal organoids for this specific variant.

Minor Points

Line 43: Do the authors mean ‘inheritance’ instead of ‘heritability’?

A: Thank you for pointing out. We have amended this to inheritance.

Line 48: Please choose between ‘~10%’ and ‘approximately 10%’.

A: Thank you for pointing out. We have amended this to ‘approximately 10%’.

Line 86: The term ‘co-transfected’ in the expression ‘then co-transfected with AONs’ means that after midigene the cells are then (doubly) transfected with a particular AON, or else that they were co-transfected with a set of AONs? Please clarify this ambiguity.

A: While performing splicing-correction assay on the HEK293T-midigene system, first we transfect confluent HEK293T cells with the midigene of choice carrying the splicing-affecting variant. We then employ a 24 hours incubation period to allow the midigene transcript to be expressed. After 24 hours, we co-transfect the particular AON. After 48 hours incubation after the co-transfection of the AON, cells are harvested for RT-PCR transcript analysis.

Line 131: Where it reads ‘Hundred microliter’ it must read ‘One hundred mL’.

A: Thank you for pointing out. We have amended this to ‘One hundred µL.’

Line 151: Where it reads ‘recombinant human COCO’ it must read ‘recombinant human COCO protein’.

A: Thank you for pointing out. We have amended this.

Line 373: Please choose between ‘fibroblasts’ and ‘cells’.

A: Thank you for pointing out. We have amended this.

Suppl. Fig. S4 legend: Please rewrite it for the sake of clarity. Ensure that all abbreviations are defined in all legends (for instance, WT, MT here and PAT, MQ in Suppl. Fig. S3).

A: Thank you for pointing out. We have amended this.

In the attachement please find a "Cover letter_Tomkiewicz et al".

Round 2

Reviewer 2 Report

The article is now deeemed publishable in its present form.